# *Nostoc edaphicum* CCNP1411 from the Baltic Sea—A New Producer of Nostocyclopeptides

**DOI:** 10.3390/md18090442

**Published:** 2020-08-26

**Authors:** Anna Fidor, Michał Grabski, Jan Gawor, Robert Gromadka, Grzegorz Węgrzyn, Hanna Mazur-Marzec

**Affiliations:** 1Division of Marine Biotechnology, Faculty of Oceanography and Geography, University of Gdańsk, Marszałka J. Piłsudskiego 46, PL-81378 Gdynia, Poland; anna.fidor@phdstud.ug.edu.pl; 2Department of Molecular Biology, University of Gdansk, Wita Stwosza 59, 80-308 Gdansk, Poland; michal.grabski@phdstud.ug.edu.pl (M.G.); grzegorz.wegrzyn@biol.ug.edu.pl (G.W.); 3DNA Sequencing and Oligonucleotide Synthesis Laboratory, Polish Academy of Sciences, Institute of Biochemistry and Biophysics, 02-106 Warsaw, Poland; gaworj@ibb.waw.pl (J.G.); robert@ibb.waw.pl (R.G.)

**Keywords:** cyanobacteria, nostocyclopeptides, *Nostoc*, *ncp* gene cluster, nonribosomal peptide synthetase

## Abstract

Nostocyclopeptides (Ncps) constitute a small class of nonribosomal peptides, exclusively produced by cyanobacteria of the genus *Nostoc*. The peptides inhibit the organic anion transporters, OATP1B3 and OATP1B1, and prevent the transport of the toxic microcystins and nodularin into hepatocytes. So far, only three structural analogues, Ncp-A1, Ncp-A2 and Ncp-M1, and their linear forms were identified in *Nostoc* strains as naturally produced cyanometabolites. In the current work, the whole genome sequence of the new Ncps producer, *N*. *edaphicum* CCNP1411 from the Baltic Sea, has been determined. The genome consists of the circular chromosome (7,733,505 bps) and five circular plasmids (from 44.5 kb to 264.8 kb). The nostocyclopeptide biosynthetic gene cluster (located between positions 7,609,981–7,643,289 bps of the chromosome) has been identified and characterized *in silico*. The LC-MS/MS analyzes of *N*. *edaphicum* CCNP1411 cell extracts prepared in aqueous methanol revealed several products of the genes. Besides the known peptides, Ncp-A1 and Ncp-A2, six other compounds putatively characterized as new noctocyclopeptide analogues were detected. This includes Ncp-E1 and E2 and their linear forms (Ncp-E1-L and E2-L), a cyclic Ncp-E3 and a linear Ncp-E4-L. Regardless of the extraction conditions, the cell contents of the linear nostocyclopeptides were found to be higher than the cyclic ones, suggesting a slow rate of the macrocyclization process.

## 1. Introduction

Secondary metabolites produced by cyanobacteria of the genus *Nostoc* (Nostocales) are characterized by a high variety of structures and biological activities [1,2,3,4,5,6]. On the basis of chemical structure, these compounds are mainly classified to peptides, polyketides, lipids, polysaccharides and alkaloids [7]. Abundantly produced cyanopeptides with anticancer, antimicrobial, antiviral and enzyme-inhibiting activity, have attracted attention of many research groups [6,8,9,10,11,12]. Some of the metabolites, such as nostocyclopeptides (Ncps) or cryptophycins are exclusively produced by the cyanobacteria of the genus *Nostoc* (Figure 1A). Ncps constitute a small class of nonribosomal peptides. Thus far, only three analogues of the compounds and their linear forms have been discovered. This includes Ncp-A1 and Ncp-A2 detected in *Nostoc* sp. ATCC53789 isolated from a lichen collected at Arron Island in Scotland [13]. The same peptides were detected in *Nostoc* sp. ASN_M, isolated from soil samples of paddy fields in the Golestan province in Iran [14] and in *Nostoc* strains from liverwort *Blasia pusilla* L. collected in Northern Norway [15]. A different analogue, Ncp-M1, was found in *Nostoc* sp. XSPORK 13A, the cyanobacterium living in symbiosis with gastropod from shallow seawaters at the Cape of Porkkala (Baltic Sea) [16].

Ncps are composed of seven residues and a unique imino linkage formed between C-terminal aldehyde and an *N*-terminal amine group of the conserved Tyr^1^ (Figure 1B) [13,16]. The presence of modified amino acid residues, e.g., 4-methylproline, homoserine and D-configured glutamine, indicated the nonribosomal biosynthetic pathway of the molecules. Genetic analysis of *Nostoc* sp. ATCC53789 revealed the presence of the 33-kb Ncp gene cluster composed of two genes, *ncpA* and *ncpB*, encoding NcpA1-A3 and NcpB1-B4 modules. These proteins catalyze the activation and incorporation of Tyr, Gly, Gln, Ile and Ser into the Ncp structure [17]. They show high similarity to NosE and NosF which take part in the biosynthesis of nostopeptolides in *Nostoc* sp. GSV224 [18]. The *ncpFGCDE* fragment of the Ncp gene cluster is involved in the synthesis of MePro (*ncpCDE*), transport (*ncpF*) and proteolysis (*ncpG*) of the peptides. The characteristic features of the Ncp enzymatic complex in *Nostoc* sp. ATCC53789 is the presence of the epimerase domain (NcpA3) responsible for D-configuration of glutamine, and the unique reductase domain at C-terminal end of NcpB which catalyze the reductive release of a linear peptide aldehyde [19,20].

The activity of Ncps have been explored [13] and their potential as antitoxins, inhibiting the transport of hepatotoxic microcystin-LR and nodularin into the rat hepatocytes through the organic anion transporter polypeptides OATP1B1/1B3 was revealed [21]. As OATP1B3 is overexpressed in some malignant tumors (e.g., colon carcinomas) [22], Ncps, as inhibitors of this transporter protein, are suggested to be promising lead compounds for new drug development. 

In our previous studies, *Nostoc edaphicum* CCNP1411 (Figure 1A) from the Baltic Sea was found to be a rich source of cyanopeptolins, the nonribosomal peptides with potent inhibitory activity against serine proteases [6]. In the current work, the potential of the strain to produce other bioactive metabolites was explored. The whole-genome sequence of *N*. *edaphicum* CCNP1411 has been determined, and the nostocyclopeptide biosynthetic gene cluster has been identified in the strain and characterized in silico for the first time. Furthermore, the new products of the Ncp gene cluster have been detected and their structures have been characterized by LC-MS/MS.

## 2. Results and Discussion

### 2.1. Analysis of N. edaphicum CCNP1411 Genome

Total DNA has been isolated from *N*. *edaphicum* CCNP1411, and the whole genome sequence has been determined. Identified replicons of *N*. *edaphicum* CCNP1411 genome consist of the circular chromosome of 7,733,505 bps, and five circular plasmids (Table 1). Within the total size of 8,316,316 bps genome (chromosome and plasmids, Figure 2), we have distinguished, according to annotation, the total number of 6957 genes from which 6458 potentially code for proteins (CDSs), 415 are classified as pseudo-genes and 84 are coding for non-translatable RNA molecules. Pseudo-genes can be divided into subcategories due to the shift in the coding frame (180), internal stop codons (77), incomplete sequence (228), or occurrence of multiple problems (63). Genes coding for functional RNAs consist of those encoding ribosomal (rRNA) (9), transporting (tRNA) (71) and regulatory noncoding (ncRNA) (4), all embedded on the chromosome. Out of total coding and pseudo-genes sequences (6873), the vast majority (5846) initiates with the ATG start codon, while GTG and TTG occur less frequently (561 and 217 times, respectively). The frequencies of stop codons were set out as follows: TAA (3455), TAG (1750), TGA (1526). Coding and pseudo-genes sequences are distributed almost equally on the leading and complementary strand, including 3408 and 3465 sequences, respectively.

### 2.2. Non-Ribosomal Peptide Synthetase (NRPS) Gene Cluster of Nostocyclopeptides

Having the whole genome sequence of *N*. *edaphicum* CCNP1411, we have analyzed in detail the non-ribosomal peptide synthetase (NRPS) cluster, containing potential genes coding for enzymes involved in the synthesis of nostocyclopeptides. To establish correct spans for non-ribosomal peptide synthetases, 35 complete nucleotide sequence clusters derived from *Cyanobacteria* phylum were aligned resulting in hits scattered around positions 2,287,143–2,323,617 and 7,609,981–7,643,289 within the *N*. *edaphicum* CCNP1411 chromosome (7.7 Mbp) (Figure 2). This method of characterization presented the overall similarity of selected spans to micropeptin (cyanopeptolin) biosynthetic gene cluster [23] and nostocyclopeptide biosynthetic gene cluster [17], respectively. To confirm these results, the antiSMASH analysis was employed resulting in confirmation of previously defined NRPS spans and adding two more regions 1,213,069–1,258,319 and 5,735,625–5,780,238, to small extent (12% and 30%, respectively) similar to anabaenopeptin gene cluster [24]. For the purpose of this study, we focused on putative nostocyclopeptide producing non-ribosomal peptide synthetase. Annotation of the selected region revealed nine putative open reading frames (ORFs), transcribed in reverse (7) and forward (2) direction. The identified cluster was arranged in a similar fashion to AY167420.1 (nostocyclopeptide biosynthetic gene cluster from *Nostoc sp*. ATCC 53789), with the exception of two ORFs (>170 bp), intersecting operon (*ncpFGCDE*) putatively encoding proteins involved in MePro assembly, efflux and hydrolysis of products of the second putative operon *ncpAB* (Figure 3).

Two sequences ORF1 (HUN01_34350) (837 bp) and ORF2 (HUN01_34355) (1107 bp) embedded on 3′ end of nostocyclopeptide gene cluster resemble *nosF* and *nosE* genes, found in the nostopeptolide (*nos*) gene cluster [18] with 96% nucleotide sequence identities in both instances, putatively encoding for zinc-dependent long-chain dehydrogenase and a Δ1-pyrroline-5-carboxylic acid reductase. Further upstream, there is an ORF3 (HUN01_34360) (798 bp) of 98% homology to unknown gene from AF204805.2 gene cluster, suggested previously to be involved in 4-methylproline biosynthesis [17,25], due to close proximity of downstream genes encompassing this reaction, but no experimental evidence was presented. Alignment of the sequence of this putative protein have shown a sequence homology, to some extent, to 4′-phosphopantetheinyl transferase, crucial for PCP aminoacyl substrate binding (Figure 4) [26]. Moreover, partially present adenylate-forming domain within ORF4 (HUN01_34365) (165 bp) belongs to the acyl- and aryl- CoA ligases family, and may putatively engage substrate for post-translational modification of the PCP domain. Facing the same direction, an ORF5 (HUN01_34370) (1605 bp)-bearing putative domain classified as transpeptidase superfamily DD-carboxypeptidase and ORF6 (HUN01_34375) (2010 bp) homologous to ABC transporter ATP-binding protein/permease may be engaged in *ncpAB* peptide product transport [27]. Neither the ORF7 (HUN01_34780) Shine–Delgarno (SD) sequence upfront translation start codon could be assigned nor the TA-like signal ~12 nucleotides upstream could be found.

The main part of the Ncp biosynthetic gene cluster is located on the forward strand comprising two large genes which nucleotide sequences are homologous over 80% to *ncpA* and *ncpB* subunits of the *ncp* cluster in *Nostoc* sp. ATCC53789 [17]. Both these genes code for proteins consisting of repetitive modules incorporating single residue into elongating peptide. ORF 8 (HUN01_34785) (11,334 bp) encompasses three of these modules, whereas ORF 9 (HUN01_34380) (14,157 bp) encodes four modules. The core of one NRPS module consists of three succeeding domains: condensation (C), adenylation (A) and peptidyl carrier protein (PCP). Moreover, adjacent to coding spans of extreme modules, two tailoring domains were found within ORF8 and ORF9 genes (Figure 5).

Alignment of nucleotide sequences to the *ncpAB* operon revealed major differences in consecutive NcpB3 and NcpB4 modules. Utilizing the selected spans conjoined with conserved domain search allowed us to distinguish and compare C, A and PCP amino-acid sequences (Figure 6). Intrinsic modules of NRPS, with an exception of NcpB3 adenylation domain sequence, were found homologous above 91%, whereas extremes have shown the biggest composition differences ranging from 13–15% to 24% in the NcpB4 adenylation domain (Figure 6).

The determination of the whole genome sequence of *N*. *edaphicum* CCNP1411 allowed us to perform analyses of genes coding for enzymes involved in the synthesis of nostocyclopeptides. The general analysis demonstrated homology of the NRPS/PKS clusters of *N*. *edaphicum* CCNP1411 to systems occurring in other cyanobacteria, however, with some differences. The non-ribosomal consensus code [28,29] allowed to recognize and predict the substrate specificities of NRPS adenylation domains: tyrosine (NcpA1), glycine (NcpA2), glutamine (NcpA3) for NcpA and isoleucine/valine (NcpB1) serine (NcpB2) 4-methylproline/proline (NcpB3) phenyloalanine/leucin/tyrosine (NcpB4) for NcpB (Table 2). This prediction was found to be in line with the structures of the Ncps detected in *N*. *edaphicum* CCNP1411.

To devolve elongating product onto subsequent condensation domain, the studied synthetase utilizes PCP domains, subunits responsible for thiolation of nascent peptide intermediates, where post-transcriptional modification of conserved serine residue shifts the state of the domain from inactive *holo* to active *apo*. Modification of this residue is related to PPTase which transfers a covalently-bound 4′-phosphopantetheine arm of CoA onto the PCP active site, enabling peptide intermediates to bind as reactive thioesters. Case residue which undergoes a nucleophilic attack by the hydroxyl group was conserved in every module within the PCP domain predicted at the front of the second helix [30].

The stand-alone docking domain (D) (7,617,812–7,617,964 bp) found on N-terminus of NcpA may be an essential component mediating interactions, recognition and specific association within NRPS subunits. The potential acceptor domain, based on sequence homology of conserved residues to C-terminal communication-mediating donor domains (COM), was found at the NcpB4 PCP domain second helix, encompassing conserved serine residue within potential binding sequence [31]. Moreover, this communication-mediating domain may putatively bind to C-terminus of NcpB3 and NcpB4 condensation (C) domains based on conserved motif LL**E**G**I**V, found by sequence homology to last five amino-acids of C-terminal docking domains residues, key for their interactions [32]. Within the same β-hairpin, a group of charged residues (ExxxxxKxR) putatively determines the binding affinity of the N-terminal domain [33].

Two tailoring domains encoded at the 5′ ends of *ncpA* and *ncpB* genes were classified as epimerization (E) (7,627,742–7,629,043 bp) domain and reductase (R) (7,642,183–7,643,238 bp) domain, accordingly. Epimerization domain catalyzes the conversion of L-amino acids to D-amino acids, a reaction coherent with D-stereochemistry of the peptide glutamine residue, where His of the conserved HHxxxDG motif and Glu from the upstream EGHGRE motif raceB comprise an epimerisation reaction active site [34]. Homologous HHxxxDG conserved motif sequence is found in condensation domains (C), where a similar reaction is catalyzed within peptide bond formation, putatively by the second His residue [35]. As in *ncp* cluster [17], module NcpA1 motif includes degenerate sequences in two positions HQIVGDL with leucine instead of phenylalanine residue at the start of the helix. The second histidine site-directed mutagenesis abolished enzymatic activity which might suggest that NcpA1 condensation domain is inactive [36].

Reductase domain (R) found at the C-terminus of NRPS was classified as oxidoreductase. Despite 15% discrepancy in domain composition compared to NcpB core catalytic triad Thr-Tyr-Lys and Rossmann-fold, a NAD (P) H nucleotide-binding motif GxxGxxG positions were not affected. The mechanism driving this chain release utilizes NAD (P) H cofactor for redox reaction of the final moiety of the nascent peptide to aldehyde or alcohol [37,38].

### 2.3. Structure Characterization of Ncps Produced by N. edaphicum CCNP1411

Thus far, only three Ncps, Ncp-A1, A2 and M1, and their linear aldehydes were isolated as pure natural products of *Nostoc* strains [13,16]. Ncp-A3, with MePhe in the C-terminal position, was obtained through aberrant biosynthesis in the *Nostoc* sp. ATCC53789 culture supplemented with MePhe [13]. The linear aldehydes of Ncp-A1 and Ncp-A2, with Pro instead of MePro, were chemically synthesized and used to study the Ncps epimerization and macrocyclization equilibria [19,20]. In our work, ten Ncps, differing mainly in position 4 and 7, were detected by LC-MS/MS in the *N*. *edaphicum* CCNP1411 cell extract (Table 3, Figure 1, Figure 7, Figure 8 and Appendix A). These include five cyclic structures, four linear Ncp aldehydes, and one linear hexapeptide Ncp. The putative structures of the six peptides, which were found to be naturally produced by *Nostoc* for the first time, are marked in Table 3 in bold (Ncp-E1, Ncp-E1-L, Ncp-E2, Ncp-E2-L, Ncp-E3 and Ncp-E4-L). 

The process of *de novo* structure elucidation was performed manually, based mainly on a series of b and y fragment ions produced by a cleavage of the peptide bonds (Figure 7, Figure 8 and Figure 9, Appendix A), and on the presence of immonium ions (e.g., *m/z* 70 for Pro, 84 for MePro, 136 for Tyr) in the product ion mass spectra of the peptides. The process of structure characterization was additionally supported by the previously published MS/MS spectra of Ncps [14]. The fragment ions that derived from the two amino acids in C-terminus usually belonged to the most intensive ions in the spectra and in this study they facilitated the structure characterization. For example, in the product ion mass spectrum of Ncp-A1 (Appendix A) and Ncp-E3 (Appendix A), ions at *m/z* 209 [MePro+Leu+H] and m/z 181 [MePro+Leu+H−CO] were present, while in the spectrum of Ncp-E2 (Appendix A) with Pro (instead MePro), the corresponding ions at 14 unit lower *m/z* values, i.e., 195 and 167 were observed. The spectra of the linear Ncps contained the intensive Tyr immonium ion at *m/z* 136. Based on the previously determined structures of Ncp-A1 and Ncp-A2 [13], we assumed that in Ncp-E2, the amino acids in position 4 and 7, are Ile and Leu, respectively (Table 3; Appendix A). These two amino acids are difficult to distinguish by MS. Definitely, the NMR analyses are required to confirm the structures of the Ncps. The presence of Val in position 4, instead of Ile, distinguishes the Ncp-E3 from other Ncps produced by *N*. *edaphicum* CCNP1411. As it was previously reported [17], and also confirmed in this study, the predicted substrates of the NcpB1 protein encoded by *ncpB* and involved in the incorporation of the residue in position 4 are Ile/Leu and Val. However, the domain preferentially activates Ile, which explains why only traces of Val-containing Ncps were detected in *N*. *edaphicum* CCNP1411 (Table 3).

Methylated Pro (MePro) in position 6 is quite conserved. MePro is a rare non-proteinogenic amino acid biosynthesized from Leu through the activity of the zinc-dependent long chain dehydrogenases and Δ^1^-pyrroline-5-carboxylic acid (P5C) reductase homologue encoded by the gene cassette *ncpCDE* [17,18,25]. The genes involved in the biosynthesis of MePro were found in 30 of the 116 tested cyanobacterial strains, majority (80%) of which belonged to the genus *Nostoc* [39]. The new Ncp-E1 and Ncp-E2, detected at trace amounts, are the only Ncps produced by *N*. *edaphicum* CCNP1411 which contain Pro (Table 3). The presence of *m/z* 84 ion in the fragmentation spectra of the two Ncps complicated the process of *de novo* structure elucidation. This ion corresponds to the immonium ion of MePro and could indicate the presence of this residue. However, the two ions *m/z* 101 and 129, which together with ion at *m/z* 84, are characteristic of Gln, suggested the presence of this amino acid in Ncp-E1 and Ncp-E2. The detailed characterization of Ncp fragmentation pathways is presented in Figure 7, Figure 8 and Figure 9 and in Appendix A. 

In addition to the heptapeptide Ncps, *N*. *edaphicum* CCNP1411 produces a small amount of the linear hexapeptide, Ncp-E4-L, whose putative structure is Tyr+Gly+Gln+Ile^+^Ser+MePro (Table 3, Figure 9). This Ncp was detected only when higher biomass of *Nostoc* was extracted. As the proposed amino acids sequence in this molecule is the same as the sequence of the first six residues in Ncp-A1 and Ncp-A2, the hexapeptide can be a precursor of the two Ncps. The other option is that the cell concentration of the Ncps is self-regulated and the Ncp-E4-L is released through proteolytic cleavage of the final products. This hypothesis could be verified when the role of the Ncps for the producer is discovered. In the *ncp* gene cluster, the presence of *ncpG* encoding the NcpG peptidase, with high homology to enzymes hydrolyzing D-amino acid-containing peptides was revealed by Becker et al. [17] and also confirmed in this study. Therefore, the in-cell degradation of Ncps by the NcpG peptidase is possible, but it probably proceeds at D-Gln and gives other products than Ncp-E4-L.

### 2.4. Production of Ncps by N. edaphicum CCNP1411

Apart from the structural analysis, we also made attempts to determine the relative amounts of the individual Ncps produced by *N*. *edaphicum* CCNP1411. To exclude the effect of the extraction procedure on the amounts of the detected peptides, different solvents and pH were applied. As the process of Ncp linearization during long storage of the freeze-dried material was suggested [16], both the fresh and lyophilized biomasses were analyzed. Regardless of the extraction procedure, Ncp-A2-L with Phe in C-terminus was always found to be the main Ncp analogue (Figure 10A–D). In addition, when MePro and Pro-containing peptide were compared separately, the peak intensity of the linear Ncps with Phe in C-terminus (i.e., Ncp-A2-L and Ncp-E1-L) was higher than the Ncps with Leu. These results might indicate preferential incorporation of Phe into the synthesized peptide chain.

The study also showed that the cell contents of the linear Ncps are higher than the cyclic ones. (Figure 10A–D and Appendix A). The release of Ncps from the synthetase as linear aldehydes is catalyzed by a reductase domain, located in the C-terminal part of the NcpB [17]. This reductive release triggers the spontaneous, and enzyme independent, macrocyclization of the linear peptide [19,20]. The reaction leads to the formation of a stable imino bond between the C-terminal aldehyde and the N-terminal amine group of Tyr [19,20]. In *N*. *edaphicum* CCNP1411 cells, depending on the Ncp analogue, the analyzed material (fresh or lyophilized) and extraction solvent, the cyclic Ncps constituted from even less than 10% (Ncp-A2, fresh biomass) to over 90% of the linear peptide (Figure 10A–D and Appendix A). In case of Ncp-A1, with MePro-Leu in C-terminus, the contribution of the cyclic form was always most significant, and at pH 8 it reached up to 91.7% of the linear peptide aldehyde (Ncp-A1-L) (Figure 10A–D and Appendix A). The cyclic analogues, Ncp-E1 and Ncp-E3 were produced in trace amounts and their spectra were sporadically recorded. It was proven that the macrocyclization process of Ncps is determined by the geometry of the linear peptide aldehyde and the conformation of D-Gln and Gly is crucial for the folding and formation of the imino bond [19]. As these two residues are present in all detected Ncps, then, probably other elements of the structure affect the cyclization process, as well. We hypothesize that due to the steric hindrances, the cyclization of Ncp-A1 with Leu in C-terminal position is easier than the cyclization of Ncp-A2 with Phe. As a consequence, the proportion of the cyclic Leu-containing Ncp-A1 to the linear form of the peptide is higher.

Thus far, Ncps synthesis was reported in few *Nostoc* strains, and the structural diversity of the peptides was found to be low. Other classes of NRPs were detected in cyanobacteria representing different orders and genera, and within one class of the peptides numerous analogues were detected. For example, the number of naturally produced cyclic heptapeptide microcystins (MCs), is over 270 [40,41] and in one cyanobacterial strains even 47 MCs analogues were detected [40]. Cyanopeptolins are produced by many cyanobacterial taxa and so far more than 190 structural analogues of the peptides have been discovered [41]. In this work, cyanopeptolin gene cluster was identified in *N*. *edaphicum* CCNP1411 and thirteen products of the genes were previously reported [6]. These peptides contain seven amino acids and a short fatty acid chain, and only one element of the structure, 3-amino-6-hydroxy-2-piperidone (Ahp), is conserved [6]. The structural diversity of NRPs is generated by frequent genetic recombination events and point mutations in the NRP gene cluster. The changes in gene sequences affect the structure and substrate specificity of the encoded enzymes. The tailoring enzymes can further modify the product, leading to even higher diversity of the synthetized peptides [42]. In case of Ncps, both the number of the producing organisms and the structural diversity of the peptides are limited. Ncp-M1 from *Nostoc* living in symbiosis with gastropod [16] is the only Ncp with structure distinctly different from Ncp-A1, Ncp-A2 and other Ncps described in this work. 

The diversity within one class of bioactive metabolites offers a good opportunity for structure-activity relationship studies, without the need to synthesize the variants. The studies are of paramount importance when the efficacy and safety of a drug candidate are optimized. Therefore, in our future work, when sufficient quantities of pure Ncps are isolated, the activity of individual analogues against different cellular targets will be tested and compared, in order to select the lead compound for further studies. 

## 3. Materials and Methods 

### 3.1. Isolation, Purification and Culturing of Nostoc CCNP1411

*Nostoc* strain CCNP1411 was isolated in 2010 from the Gulf of Gdańsk, southern Baltic Sea, by Dr. Justyna Kobos. Based on the 16S rRNA sequence (GenBank accession number KJ161445) and morphological features, such as the shape of trichomes, cell size (4.56 ± 0.30 µm wide and 4.12 ± 0.72 µm long) and lack of akinetes [43,44], the strain was classified to *N*. *edaphicum* species. Purification of the strain was carried out by multiple transfers to a liquid and solid (1% bacterial agar) Z8 medium supplemented with NaCl to obtain the salinity of 7.3 [45]. To establish the strain as a monoculture, free from accompanying heterotrophic bacteria, a third-generation cephalosporin, ceftriaxone (100 µg/mL) (Pol-Aura, Olsztyn, Poland) was used. In addition, the purity of the culture was regularly tested by inoculation on LA agar (solid LB medium with 1.5% agar) [46] and on agar Columbia +5% sheep blood (BTL Ltd. Łódź, Poland), a highly nutritious medium, recommended for fastidious bacteria. Cyanobacteria cultures were grown in liquid Z8 medium (100 mL) at 22 ± 1 °C, continuous light of 5–10 µmol photons m^−2^ s^−1^. After three weeks of growth, the cyanobacterial biomass was harvested by passing the culture through a nylon net (mesh size 25 µm) and then freeze-dried before further processing.

### 3.2. Isolation and Sequencing of Genomic DNA 

Genomic DNA of *N*. *edaphicum* CCNP1411 was isolated using SDS/Phenol method as described previously [47,48]. DNA quality control was performed by measuring the absorbance at 260/230 nm, template concentration was determined using Qubit fluorimeter (Thermo Fisher Scientific, Waltham, MA, USA), and DNA integrity was analyzed by 0.8% agarose gel electrophoresis and by PFGE using Biorad CHEF-III instrument (BioRad, Hercules, CA, USA).

Paired-end sequencing library was constructed using the NEB Ultra II FS Preparation Kit (New England Biolabs, Beverly, CA, USA) according to the manufacturer’s instructions. The library was sequenced using an Illumina MiSeq platform (Illumina, San Diego, CA, USA) with 2 × 300 paired-end reads. Sequence quality metrics were assessed using FASTQC (http://www.bioinformatics.babraham.ac.uk/projects/fastqc/) [49].

The long reads were obtained using the GridION sequencer (Oxford Nanopore Technologies, Oxford, UK). Prior to long-read library preparation, genomic DNA was sheared into 30 kb fragments using 26 G needle followed by size selection on Bluepippin instrument (Sage Science, Beverly, MA, USA). DNA fragments above 20 kb were recovered using PAC30 kb cassette. 5 µg of recovered DNA was taken for 1D library construction using SQK-LSK109 kit and 0.5 µg of the final library was loaded into R9.4.1 flowcell and sequenced on MinION sequencer. 

### 3.3. Genome Assembling 

Raw nanopore data was basecalled using Guppy v3.2.2 (Oxford Nanopore Technologies, Oxford, UK). After quality filtering using NanoFilt [50] and residual adapter removal using Porechop (https://github.com/rrwick/Porechop), the obtained dataset was quality checked using NanoPlot [50]. Long nanopore reads were then assembled using Flye v2.6 [51]. Flye assembled contigs were further polished using Illumina sequencing reads and Unicycler_polish pipeline (https://github.com/rrwick/Unicycler).

### 3.4. Genome and NRPS Alignment

Genome assembly was annotated using the NCBI Prokaryotic Genome Annotation Pipeline [52] with the assistance of prokka [53] refine annotation, with additionally curated database comprised of sequences selected by Nostocales order from NCBI non-redundant and refseq_genomes (280 positions) databases, enriched by 35 NRPS/PKS clusters selected by cyanobacteria phylum. To create circular maps of *N*. *edaphicum* CCNP1411 genome, the CGView Comparison Tool [54] was engaged with additional GC skew and GC content analyses.

Selected span for potential NRPS cluster was confirmed with BLASTn, BLASTp [55] and antiSMASH [56]. ORFs start codons within a putative cluster were verified by the presence of ribosome binding sites, 4–12 nucleotides upstream of the start codon. Schematic comparison of ORF BLASTn from relative synthetases, AY167420.1 and CP026681.1, was visualized by EasyFig program (http://mjsull.github.io/Easyfig/files.html). Annotated regions of NRPS span were subjected for NCBI Conserved Domain Database search [57] with a set e-value threshold (10^−3^), determining evolutionarily-conserved protein domains and motifs against CDD v3.18 database. Recognized motifs were selected using samtools v.1.9 and were subjected for protein structure and function prediction by I-TASSER [58], and results were confirmed with literature reports, PKS/NRPS Analysis Web-site prediction [59] and reevaluated using MEGAX suite [60]. Amino-acid sequence was visualized by BOXSHADE 3.2 program (https://embnet.vital-it.ch/software/BOX_form.html). Determination of domain ligand binding and active sites was achieved using COFACTOR and COACH part of I-TASSER analyses confirmed by MUSCLE amino acid alignment from MEGA X.

### 3.5. Data Deposition

Genomic sequences generated and analyzed in this study were deposited in the GenBank database under BioProject number: PRJNA638531.

### 3.6. Extraction and LC-MS/MS Analysis

For LC-MS/MS analyses of Ncps, the lyophilized (10 mg DW) biomass of *N*. *edaphicum* CCNP1411 was homogenized by grinding with mortar and pestle, and extracted in 1 mL of milliQ water, 20% methanol (pH 3.5, 6.0 and 8.0) and 50% methanol in water. The pH was adjusted with 0.5 M HCl and 1.0 M NaOH. In addition, the fresh material (500 mg FW) was extracted in 20% methanol in water. The samples were vortexed for 15 min and centrifuged at 14,000 rpm for 10 min, at 4 °C. The collected supernatants were directly analyzed by LC-MS/MS system.

The LC-MS/MS was carried out on an Agilent 1200 HPLC (Agilent Technologies, Waldbronn, Germany) coupled to a hybrid triple quadrupole/linear ion trap mass spectrometer QTRAP5500 (Applied Biosystems MDS Sciex, Concord, ON, Canada). The separation was achieved on a Zorbax Eclipse XDB-C18 column (4.6 mm ID × 150 mm, 5 µm; Agilent Technologies, Santa Clara, CA, USA). The extract components were separated by gradient elution from 10% to 100% B (acetonitrile with 0.1% formic acid) over 25 min, at a flow rate of 0.6 mL/min. As solvent A, 5% acetonitrile in MilliQ water with 0.1% formic acid was used. The mass spectrometer was operated in positive mode, with turbo ion source (5.5 kV; 550 °C). An information-dependent acquisition method at the following settings was used: for ions within the *m/z* range 500–1100 and signal intensity above the threshold of 500,000 cps the MS/MS spectra were acquired within the *m/z* range 50–1000, at a collision energy of 60 eV and declustering potential of 80 eV. Data were acquired with the Analyst ^®^ Sofware (version 1.7 Applied Biosystems, Concord, ON, Canada).

## 4. Conclusions

Genes coding for subunits of the non-ribosomal peptide synthetase, in nostocyclopeptide-producing *N*. *edaphicum* CCNP1411, revealed differences in nucleotide compositions, compared to the previously described *ncp* cluster of *Nostoc* sp. ATCC53789. Although the analysis of fragments of genes coding for active sites and ligand binding sites of the conserved protein domains derived from *N*. *edaphicum* CCNP1411 and *Nostoc* sp. ATCC53789 indicated identical amino-acid compositions, residues within adenylation domains and substrate binding sites were different between compared sequences. This finding may highlight sites prone to mutations within regions accounted for structure and substrate stability. Analysis of *ncp* gene products in *N*. *edaphicum* CCNP1411 led to the detection of new nostocyclopeptide analogues. However, modifications in their structure were minor and limited to three positions of the heptapeptides. Although the naturally produced nostocyclopeptides were previously described as cyclic structures, in *N*. *edaphicum* CCNP1411 they are mainly present as linear peptide aldehydes, indicating a slow cyclization process.

## Figures and Tables

**Figure 1 marinedrugs-18-00442-f001:**
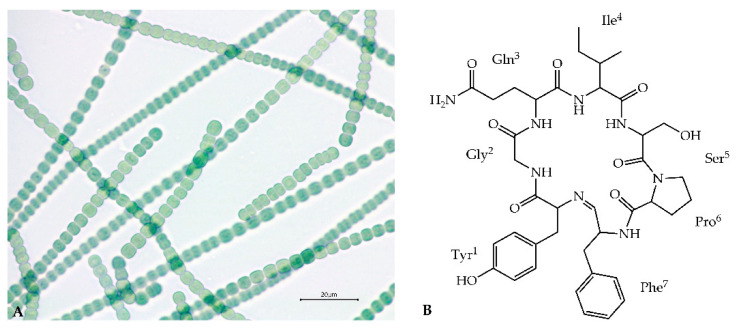
*Nostoc edaphicum* CCNP1411 (**A**) and the proposed chemical structure of Ncp-E1 (**B**).

**Figure 2 marinedrugs-18-00442-f002:**
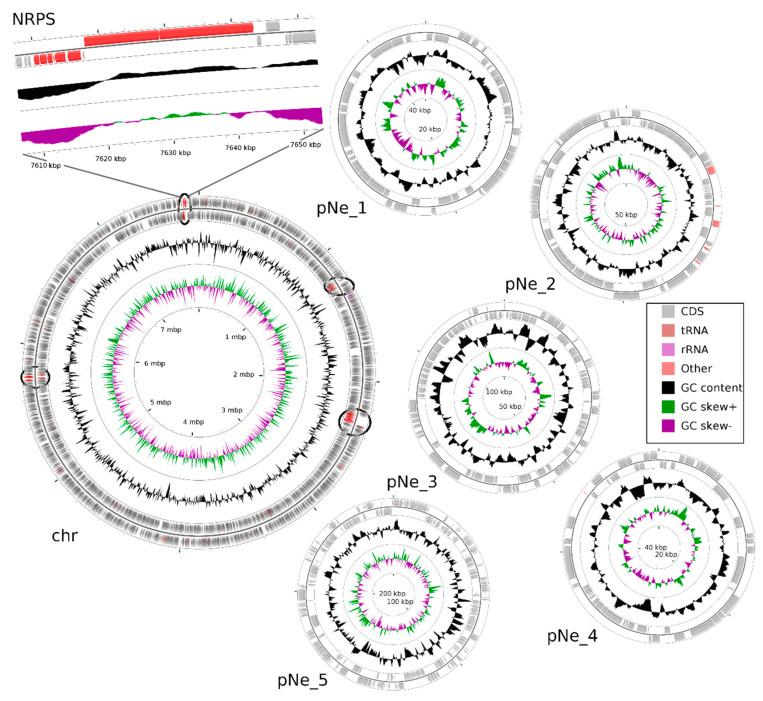
Map of the *N*. *edaphicum* CCNP1411 genome where chromosome (chr) and five plasmids (pNe_1–5) are presented. The ORFs are indicated with grey arrows split into two rings outermost showing ORFs on direct strand and inner showing complementary strand ORFs. The middle circle shows GC content (black) and the innermost circle shows GC skew (green and purple). Genes for putative NRPS/PKS are marked on the chromosome in their proper position (red within a black circle), with closeup on NRPS in position 7,609,981–7,643,289 putatively coding for Ncp biosynthetic gene cluster.

**Figure 3 marinedrugs-18-00442-f003:**
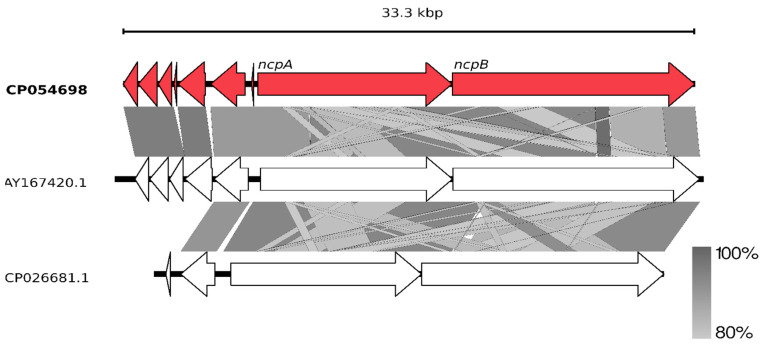
Schematic alignment of genes coding for putative non-ribosomal peptide synthetase from *N*. *edaphicum* CCNP1411 (red) and two related Ncp-producing synthetases AY167420.1 and CP026681.1 (white). The grey bar in the lower right corner shows the identity percentage associated with color of the bars connecting homologous regions. The analysis was conducted at the nucleotide level.

**Figure 4 marinedrugs-18-00442-f004:**
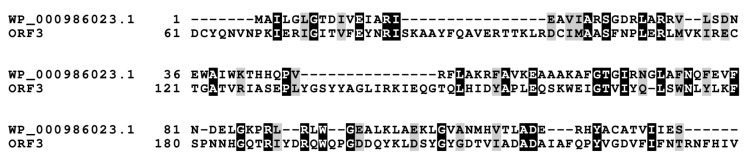
Structure-based sequence alignment of 4′-phosphopantetheinyl transferase and partial ORF3. Amino-acids highlighted in black color indicate conserved residues, whereas those in grey color indicate conservative mutations.

**Figure 5 marinedrugs-18-00442-f005:**
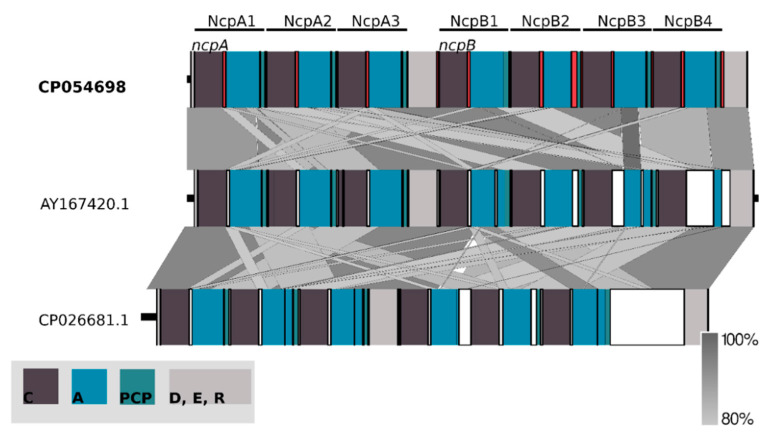
Schematic representation of conserved domains within *ncpA* and *ncpB* coding nucleotide sequences. They are composed of repetitive modules condensation (C), adenylation (A) and peptidyl carrier protein (PCP) domains adjacent to delineating docking, epimerization and reductase domains aligned with two related synthetases AY167420.1 and CP026681.1. The analysis was conducted at the nucleotide level.

**Figure 6 marinedrugs-18-00442-f006:**
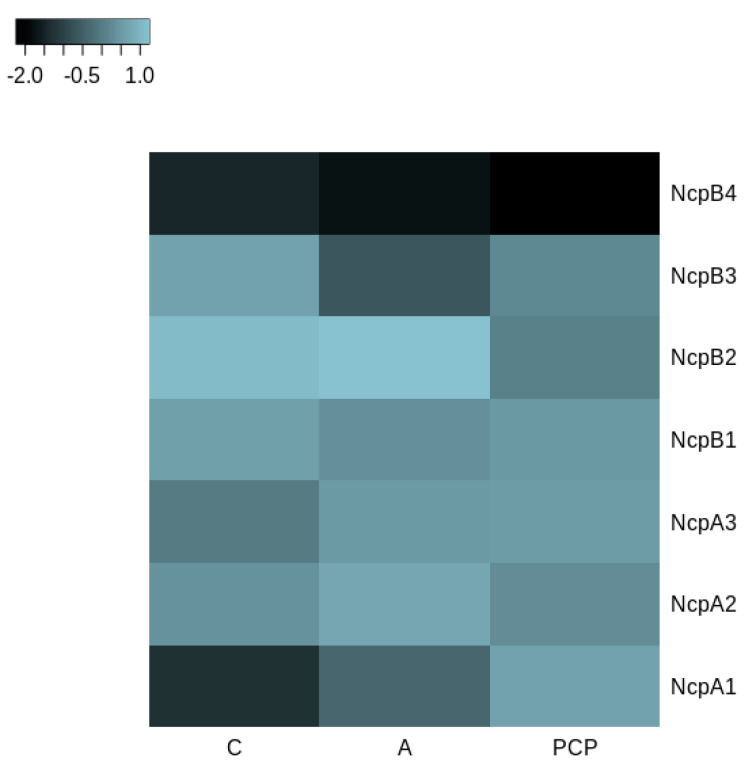
Heatmap of the highest (light blue) and lowest (black) percentage of similarities between NcpA and NcpB domains in *N*. *edaphicum* CCNP1411 and ATCC53789; values scaled by rows. The analysis was conducted at the amino acid level.

**Figure 7 marinedrugs-18-00442-f007:**
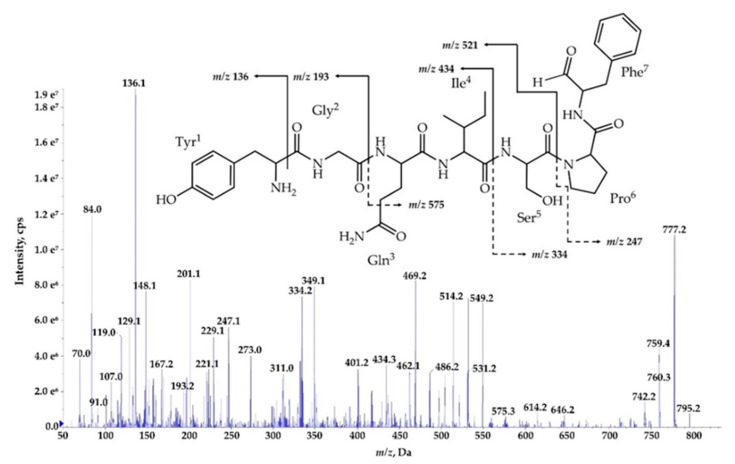
Postulated structure and enhanced product ion mass spectrum of the linear aldehyde nostocyclopeptide **Ncp-E1-L;** Tyr+Gly+Gln+Ile+Ser+Pro+Phe characterized based on the following fragment ions: *m/z* 795 [M+H]; 777 [M+H−H_2_O]; 759 [M+H−2H_2_O]; 646 [M+H – Phe]; 614 [M+H – Tyr−H_2_O]; 575 [M+H−(Tyr+Gly)]; 549 [Tyr+Gly+Gln+Ile+Ser+H]; 531 [Tyr+Gly+Gln+Ile+Ser+H−H_2_O]; 462 [Tyr+Gly+Gln+Ile+H]; 349 [Tyr+Gly+Gln+H]; 334 [Ser+Pro+Phe+2H]; 247 [Phe+Pro+H]; 229 [Phe+Pro+H]; 201 [Phe+Pro+H – CO]; 148 [Tyr−NH_2_]; 136 Tyr immonium; 129, 101 (immonium), 84 Gln; 70 Pro immonium.

**Figure 8 marinedrugs-18-00442-f008:**
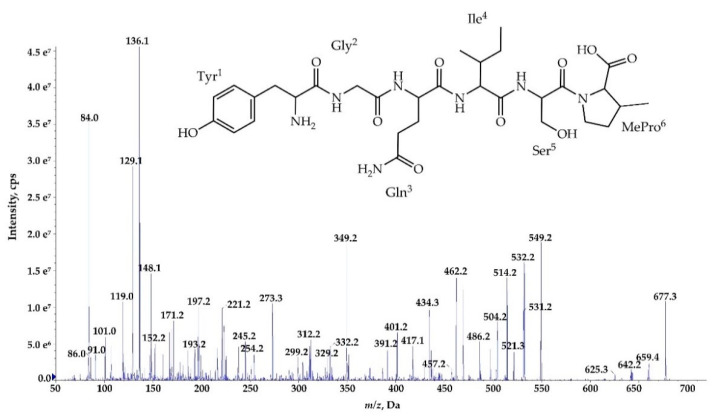
Postulated tructure and enhanced product ion mass spectrum of a linear nostocyclopeptide **Ncp-E4-L** [Tyr+Gly+Gln+Ile+Ser+MePro] characterized based on the following fragment ions: *m/z* 677 [M+H]; 659 [M+H−H_2_O]; 642 [M+H−H_2_O−NH_3_]; 549 [Tyr+Gly+Gln+Ile+Ser+H]; 531 [Tyr+Gly+Gln+Ile+Ser+H−H_2_O]; 521 [Tyr+Gly+Gln+Ile+Ser+H−CO]; 462 [Tyr+Gly+Gln+Ile+H]; 434 [Tyr+Gly+Gln+Ile+H−CO]; 349 [Tyr+Gly+Gln+H]; 329 [Gln+Ile+Ser+H]; 312 [Ile+Ser+MePro+H]; 221 [Tyr+Gly+H]; 193 [Tyr+Gly+H−CO]; 148 [Tyr−NH_2_]; 136 Tyr immonium; 86 Ile immonium; 84, 101 (immonium), 129 Gln; 84 MePro immonium.

**Figure 9 marinedrugs-18-00442-f009:**
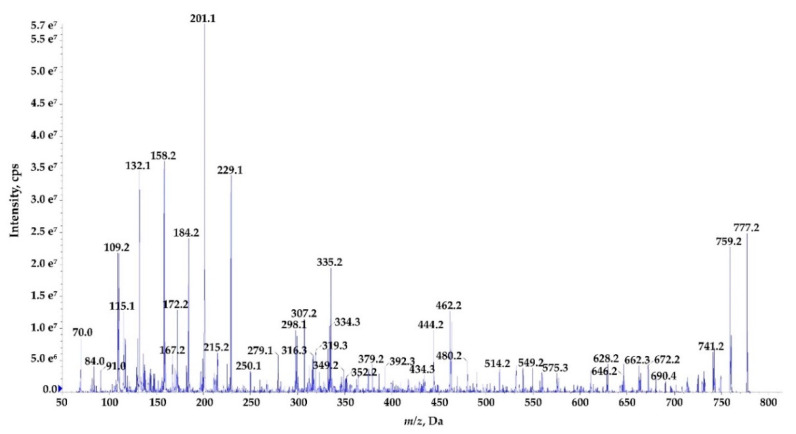
Enhanced product ion mass spectrum of the cyclic nostocyclopeptide **Ncp-E1** with putative structure cyclo[Tyr+Gly+Gln+Ile+Ser+Pro+Phe] characterised based on the following fragment ions: *m/z* 777 [M+H]; 759 [M+H−H_2_O]; 741 [M+H−2H_2_O]; 690 [M+H−Ser]; 672 [M+H−Ser−H_2_O]; 662 [M+H−Pro−H_2_O]; 646 [M+H−Phe]; 628 [M+H−Phe−H_2_O]; 575 [M+H−(Ser+Pro)−H_2_O]; 549 [Tyr+Gly+Glu+Ile+Ser+H]; 480 [Phe+Tyr+Gly+Gln+H]; 462 [Tyr+Gly+Gln+Ile+H]; 444 [Tyr+Gly+Gln+Ile+H−H_2_O]; 434 [Tyr+Gly+Gln+Ile+H−CO]; 392 [Pro+Phe+Tyr+H]; 352 [Phe+Tyr+Gly+H]; 335 [Phe+Tyr+Gly+H−H_2_O]; 316 [Ser+Pro+Phe+H]; 307 [Phe+Tyr+Gly+H−H_2_O -CO]; 298 [Ile+Ser+Pro+H]; 229 [Phe+Pro+H]; 201 [Phe+Pro+H−CO]; 158 [Gly+Gln+H−CO]; 132 Phe; 70 Pro immonium. Structure of the peptide is presented in Figure 1.

**Figure 10 marinedrugs-18-00442-f010:**
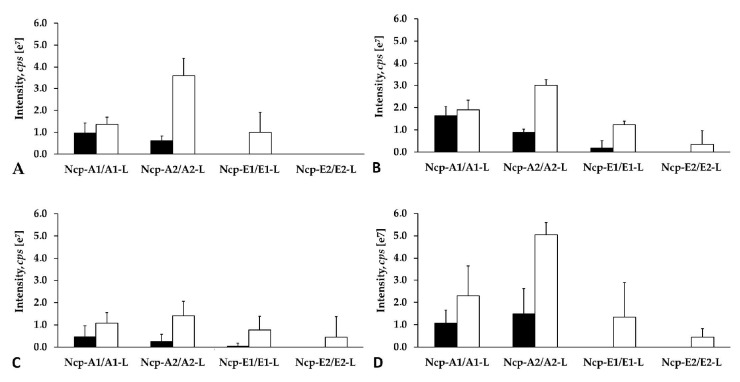
Relative cell contents of nostocyclopeptides extracted from 10 mg of lyophilized (**A**,**B**,**D**) or 500 mg fresh (**C**) biomass of *N*. *edaphicum* CCNP1411 with MilliQ water (**A**), 50% methanol in water (**B**), and 20% methanol in water (**C**,**D**). The cell content was expressed as peak intensity in LC-MS/MS chromatogram.

**Table 1 marinedrugs-18-00442-t001:** Composition and coverage of *N*. *edaphicum* CCNP1411 genome.

Replicon	Accession Number	Length (bp)	Topology	G+C Content (%)	Coverage (x) Nanopore Data	Coverage (x) Illumina Data
pNe_1	CP054693.1	44,503	Circular	42.3	115.5	244.8
pNe_2	CP054694.1	99,098	Circular	40.2	168.4	135.3
pNe_3	CP054695.1	120,515	Circular	41.3	256.4	177.1
pNe_4	CP054696.1	53,840	Circular	41.6	102.4	211.5
pNe_5	CP054697.1	264,855	Circular	41.0	226.3	160.7
chr	CP054698.1	7,733,505	Circular	41.6	160.7	116.9

**Table 2 marinedrugs-18-00442-t002:** Characterization of substrate binding pocket amino acid residues adenylation domains of NcpA and NcpB modules based on gramicidin S synthetase (GrsA) phenylalanine activating domain. Residues in brackets mark inconsistency with AY167420.1 residues.

NRPS Module	Adenylation Domain Residue Position									Proposed Substrate
	235	236	239	278	299	301	322	330	331	
**NcpA1**	D	A	S	T	[I]	A	A	V	C	Tyr
NcpA2	D	I	L	Q	L	G	L	I	W	Gly
NcpA3	D	A	W	Q	F	G	L	I	D	Gln
NcpB1	D	A	F	F	L	G	V	T	F	Ile/Val
NcpB2	D	V	W	H	I	S	L	I	D	Ser
NcpB3	D	V	Q	[F]	I	A	H	V	A	Pro/MePro
NcpB4	D	A	W	[T]	I	G	[A]	V	C	Phe/Tyr/Leu

**Table 3 marinedrugs-18-00442-t003:** The putative structures of nostocyclopeptides (Ncps) detected in the crude extract of *N*. *edaphicum* CCNP1411 and the structure of Ncp-M1 identified in *Nostoc* sp. XSPORK 13 A [16]. The new analogues are marked in bold and the peptides detected in trace amounts are marked with [T]. The variable residues in position 4 and 7 are marked in blue.

Compound	Structure	*m/z* [M+H]^+^		Retention Time [min]
		Cyclic	Linear–COH	
**Ncp-A1**	**cyclo[Tyr+Gly+Gln+Ile+Ser+MePro+Leu]**	757		7.1
Ncp-A1-L	Tyr+Gly+Gln+Ile+Ser+MePro+Leu		775	5.8
Ncp-A2	cyclo[Tyr+Gly+Gln+Ile+Ser+MePro+Phe]	791		6.0
Ncp-A2-L	Tyr+Gly+Gln+Ile+Ser+MePro+Phe		809	5.6
**Ncp-E1**	**cyclo[Tyr+Gly+Gln+Ile+Ser+Pro+Phe]**	**777 [T]**		**7.2**
**Ncp-E1-L**	**Tyr+Gly+Gln+Ile+Ser+Pro+Phe**		**795**	**5.7**
**Ncp-E2**	**cyclo[Tyr+Gly+Gln+Ile+Ser+Pro+Leu]**	**743 [T]**		**6.3**
**Ncp-E2-L**	**Tyr+Gly+Gln+Ile+Ser+Pro+Leu**		**761**	**5.1**
**Ncp-E3**	**cyclo[Tyr+Gly+Gln+Val+Ser+MePro+Leu]**	**743 [T]**		**7.0**
*** Ncp-E4-L**	**[Tyr+Gly+Gln+Ile+Ser+MePro]**		**677 [T]**	**6.0**
** Ncp-M1	cyclo[Tyr+Tyr+HSe+Pro+Val+MePro+Tyr]		882	27.5

* Ncp-E4-L is the only linear Ncps analogue with carboxyl group in C-terminus. ** Identified in *Nostoc* sp. XSPORK [16].

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
