# Peer review of "Nostoc edaphicum CCNP1411 from the Baltic Sea—A New Producer of Nostocyclopeptides"

_marinedrugs, 2020, doi:10.3390/md18090442_

Round 1

Reviewer 1 Report

Review comments on “Nostoc edaphicum CCNP 1411 from the Baltic Sea–a new producer of nostocyclopeptides”

General comment:

Be consistent in using the strain number as “CCNP1411” and not as “CCNP 1411” throughout the manuscript.

Although the partial 16S rRNA gene sequence of this organism is available in GenBank, it appears no morpho-taxonomical information of this newly isolated cyanobacterium was published. If this is true, then for the benefit of cyanobacteriologists, it would be nice if you add a photomicrograph of this N. edaphicum in the figure 1 along with high-quality chemical structures of Ncps isolated from this organism. Also, if possible, a brief morphological description to identify this cyanobacterium may be provided.

Specific comments:

Lines21-22: “The nostocyclopeptide biosynthetic gene cluster has been identified and characterized in silico.” Mention where this ncp gene cluster was found and specify the position in the genome.

Lines22-23: “The LC-MS/MS analysis of N. edaphicum CCNP1411 revealed several products of the genes.” It is not clear from the sentence what was analysed by LC-MS/MS. Was it water soluble extract/fraction or lipid soluble extract or anything else?

Figure2: It would be great, if possible, provide high-quality images of the plasmids and chromosome maps.

Table1:Add a column to the table to provide the specific GenBank numbers for each plasmids and chromosome.

Figure legend 5: Write the abbreviated letters in the parenthesis in appropriate place in the sentence. For example, …condensation (C), adenylation (A)….

Line 404: Move the section for “3.5. Culture Conditions and Purification of Nostoc CCNP1411” prior to the section “3.1. Isolation and sequencing of genomic DNA”. Also, change the title to “Isolation, purification and culturing of Nostoc edaphicum CCNP1411”

Line 405: mention that this “KJ161445” number is related to the 16S rRNA gene of this organism.

Line 407: Provide the reference for Z8S medium.

Line 408: what concentration of “cephalosporin” was used and under what incubation conditions?

Line 410-411: Was there any rationale for using “agar Columbia +5% 410 KB” and what is the full form of “KB”

Lines 411-412: Provide the growth medium used to cultivate the Nostoc sp. with reference. At what culture volume it was cultivated? and how long before the biomass was harvested for extraction of Ncps? Briefly mention, how the biomass was harvested. Use the light intensity unit as “µmol photons m-2 s-1” which is nowadays considered as microbiological standard.

Line415: How the DW biomass was homogenised? How the pH of 20% methanol was set?

Line  428: Check if it was “m/z range 500–1100” or “m/z range 50–1100”?

Line433: Look for the typographical error?

Author Response

Review comments on “Nostoc edaphicum CCNP 1411 from the Baltic Sea–a new producer of nostocyclopeptides”

General comment:

Be consistent in using the strain number as “CCNP1411” and not as “CCNP 1411” throughout the manuscript.

Corrections have been made in the whole text of the re-submitted manuscript, including the title. The strain number is CCNP1411 (without spaces).

Although the partial 16S rRNA gene sequence of this organism is available in GenBank, it appears no morpho-taxonomical information of this newly isolated cyanobacterium was published. If this is true, then for the benefit of cyanobacteriologists, it would be nice if you add a photomicrograph of this N. edaphicum in the figure 1 along with high-quality chemical structures of Ncps isolated from this organism. Also, if possible, a brief morphological description to identify this cyanobacterium may be provided.

As suggested, the photomicrograph of the analyzed strain was added to Figure 1. A brief description of morphological features that, together with the 16S rRNA sequences, supported the taxonomic classification of the strain was added in material and methods (section 3.1).

Specific comments:

Lines21-22: “The nostocyclopeptide biosynthetic gene cluster has been identified and characterized in silico.” Mention where this ncp gene cluster was found and specify the position in the genome.

The sentence was rephrased as follows: “The genome consists of the circular chromosome (7,733,505 bps) and five circular plasmids (from 44.5 kb to 264.8 kb). The nostocyclopeptide biosynthetic gene cluster (located between positions 7,609,981-7,643,289 bps of the chromosome) has been identified and characterized in silico”.

Lines22-23: “The LC-MS/MS analysis of N. edaphicum CCNP1411 revealed several products of the genes.” It is not clear from the sentence what was analyzed by LC-MS/MS. Was it water soluble extract/fraction or lipid soluble extract or anything else?

The current form of the sentence is as follows: “The LC-MS/MS analyzes of N. edaphicum CCNP1411 cell extracts prepared in aqueous methanol revealed several products of the genes.”

Further details on the solvents used for extraction are described in the experimental section.

 Figure2: It would be great, if possible, provide high-quality images of the plasmids and chromosome maps.

Figure 2 was replaced with the improved image presenting chromosome and plasmids of N. edaphicum CCNP1411.

Table1: Add a column to the table to provide the specific GenBank numbers for each plasmids and chromosome.

In Table 1 a column was added with the GenBank accession numbers of chromosome and plasmids.

Figure legend 5: Write the abbreviated letters in the parenthesis in appropriate place in the sentence. For example, …condensation (C), adenylation (A)….

The abbreviated letters were added in the parenthesis.

Line 404: Move the section for “3.5. Culture Conditions and Purification of Nostoc CCNP1411” prior to the section “3.1. Isolation and sequencing of genomic DNA”. Also, change the title to “Isolation, purification and culturing of Nostoc edaphicum CCNP1411”

The suggested changes in the sequence of the sections and in the title of the section were made.

Line 405: mention that this “KJ161445” number is related to the 16S rRNA gene of this organism.

The information was added: “Nostoc strain CCNP1411 was isolated in 2010 from the Gulf of Gdańsk, southern Baltic Sea, by Dr. Justyna Kobos. Based on the 16S rRNA sequence (GenBank accession number KJ161445) and morphological features, such as shape of trichomes, cell size (4.56±0.30 µm wide and 4.12±0.72 µm long) and lack of akinetes, the strain was classified to N. edaphicum species.”

Line 407: Provide the reference for Z8S medium.

The reference and additional information on Z8S medium were added.

Line 408: what concentration of “cephalosporin” was used and under what incubation conditions?

The concentration of cephalosporin (ceftriaxone) was added (100 mg/mL).

Line 410-411: Was there any rationale for using “agar Columbia +5% KB” and what is the full form of “KB”

KB was changed into “sheep blood”. Agar Columbia +5% sheep blood is a rich medium recommended for fastidious bacteria. Our idea was to use medium supporting the growth of a possible broad range of bacteria – to check the Nostoc culture for contamination. The current form of the sentence is as follows:

“In addition, the purity of the culture was regularly tested by inoculation on LA agar (solid LB medium with 1.5% agar) [46] and on agar Columbia +5% sheep blood (BTL Ltd. Łódź, Poland), a highly nutritious medium, recommended for fastidious bacteria.”

Lines 411-412: Provide the growth medium used to cultivate the Nostoc sp. with reference. At what culture volume it was cultivated? and how long before the biomass was harvested for extraction of Ncps? Briefly mention, how the biomass was harvested. Use the light intensity unit as “µmol photons m-2 s-1” which is nowadays considered as microbiological standard.

The information regarding Nostoc growth medium (with reference), culture volume, time of cyanobacteria growth, way of biomass harvesting was added. The light intensity unit was corrected. The current text is as follows:

“Cyanobacteria cultures were grown in liquid Z8 medium (100 mL) at 22±1 ºC, continuous light of 5-10 µmol photons m-2 s-1. After three weeks of growth, the cyanobacterial biomass was harvested by passing the culture through a nylon net (mesh size 25 µm) and then, freeze-dried before further processing.”

Line415: How the DW biomass was homogenised? How the pH of 20% methanol was set?

The information was added in the following sentences:

“For LC-MS/MS analyses of Ncps, the lyophilized (10 mg DW) biomass of N. edaphicum CCNP1411 was homogenized by grinding with mortar and pestle, and extracted in 1 mL of milliQ water, 20% methanol (pH 3.5, 6.0 and 8.0) and 50% methanol in water. The pH was adjusted with 0.5 M HCl and 1.0 M NaOH”.

Line  428: Check if it was “m/z range 500–1100” or “m/z range 50–1100”?

The raged of ions to be fragmented was 500-1100, as specified in the text. The product ion mass spectra were registered within a range m/z 50-1000. For the sake of clarity, the latter information was added to the text of section 3.6.

“An Information Dependent Acquisition method at the following settings was used: for ions within the m/z range 500–1100 and signal intensity above the threshold of 500,000 cps, the MS/MS spectra were acquired within the m/z range 50-1000, at collision energy of 60 eV and declustering potential of 80 eV.”

Line433: Look for the typographical error?

The typographical error in the word ‘coding’ was corrected.

We thank the Reviewer for all valuable comments included in the review.

Reviewer 2 Report

The manuscript entitled as “Nostoc edaphicum CCNP 1411 from the Baltic Sea–a new producer of nostocyclopeptides” presents a study on the production of nostocyclopeptides by a Nostoc edaphicum strain isolated from the Baltic Sea.

The strengths of this work are the combination of whole genome sequencing to characterize the biosynthetic pathway as well as the putative products along with the quantification and LC-MS/MS analysis of the Ncps.

I consider that, due to the high quality of results and the interesting discussion proposed by the authors, this work is suitable for publication in the present form.

Author Response

Thank you for such positive evaluation of our work.

Reviewer 3 Report

Dear authors

The paper describes a cyanobacteria strain CCNP1411 producing new nostocyclopeptides. The DNA analysis of this strain has been done and Ncps metabolites were identified by LC-MS/MS in the crude extract. I cannot judge on the molecular biology part, but can give you some comment on the other part, hoping this can help improve the manuscript:

Here are my comments:

Experimental design

  • I don’t see the relevance of determining the relative amounts of Ncps in different extracts? I would rather try to increase the total yield in order to scale-up the extraction and isolation of these metabolites?!? Detecting new compounds are ok, but these are only preliminary results. The most important is still to isolate compounds and test. I guess as I read your last paragraph L350, you would agree, isn’t it? Your experimental design (experiment L301) doesn’t match your overall goal. If this is not relevant, I suggest removing that part.

General draft.

  • Please be rigorous with terminology used. The terminology ‘isolated’, ‘detected’ or ‘discovered’ should be used in a clear way. When you mentioning a compound, it is not always clear if you are talking about a compound that was detected (compound detected by LC-MS in a crude extract) or isolated (you have a pure compound in a vial and you can acquire NMR and test for biological activity). This should be clarified everywhere in the manuscript (from abstract, in figures and unto the conclusion).

I give you one example: in figure 1 where you show Ncp-E1 and Ncp-E1-L. You mention that these were isolated from CCNP 1411. To my understanding, these were only detected by LC-MS in a crude extract.

Abstract.

  • Detected as naturally produced cyanometabolites. What do you mean? Do you mean that only three Ncps (Ncp-A1, Ncp-A2 and Ncp-M) were isolated as pure natural products from cyanobacteria? It would be nice to have this clearly written somewhere.
  • Please replace the ‘four new variants’ with ‘four new analogues’ or ‘four new Ncps’. And it would be nice to add their code? Which are the new Ncps you have identified in that strain. Please mention as well if they were detected as trace-amount. Information should be clearly presented to the reader.

Introduction.

  • This sentence is not clear to me. You say that nostocyclopeptides were identified only in Nostoc sp. then you mention only Ncp_A1, Ncp_A2, and Ncp_M1. What about Ncp_B1-B4 for example? Were these compounds never isolated and only identified by LCMS and/or through the gene clusters analysis? The term ‘discovered’ is also not clear here. Maybe you wanted to say that only three Ncp namely Ncp-A1, -A2 and -M were isolated from Nostoc sp. All other analogues were identified by LCMS…etc. Information should be clear!!
  • You review the activity of nostocyclopeptide. Is this relevant in this paper? This is not a review, but an introduction of your work where you are even not testing any compound. This paragraph could be shortened. I would suggest to keep on the red line.
  • You mention that you report in this paper the production of nostocyclopeptides by Nostoc edaphicum. We still don’t understand why you are doing this work? what is your strategy behind? Are you studying these microbes? Are you looking for new cynobacteria? Are you sequencing all strains you isolate? Your strategy is not clear to me. I would appreciate 1-2 sentences about your strategy so the reader can understand why you are now reporting this strain. Maybe you could sell it in the previous paragraph (this is only a suggestion based on my understanding): nostocyclopeptides are displaying interesting activities such as xyz to mention just a few -> our goal was to isolated cyanobacteria from the Baltic sea -> Nostoc edaphicum was identified….
  • Is Nostoc edaphicum a new strain? or is that strain already known and the Ncps gene cluster has never been investigated?
  • These last sentences should be part of the conclusion. This has nothing to do in the introduction. In the introduction, you should only give information to help the reader understand why you did this work, why is this important.

Figures.

  • Figure 1. Please move into results. These structures have nothing to do in introduction. Introduction is not a resumé! Structures should go in Results.
  • The legend of that figure mention that these two compounds were isolated from CCNP1411. However, in method, I didn’t find the description of the isolation. Please be clear.

Results and discussion.

  • L233-L272. Could you please double check the structure elucidation part and make sure the info is clearly presented and precise. The information is often confusing.
  • You are talking about data in suppl info. So please, mention it.
  • As you are discussing some fragments, would it be possible to highlight these typical fragments in the spectra (show 1 or 2 most relevant fragments next to peak). It would help the reader follow your discussion.
  • Could you please clarify which compounds are new? In the abstract you are talking about 4 new Ncps and in table 3, I have a list of 10 Ncps.
  • Moreover, which Ncps were present in trace amount? This should be clear from the beginning.
  • Table 3. Structures. At the beginning of paragraph, would it be possible to highlight the difference between structures, so the reader can quickly understand it. I had to look carefully at the table to see that there is only a variation of 2 aa between the structures.
  • L219-220. Thus far, only three naturally produced Ncp variants …. This sentence is not clear. I would suggest to modify as follow: Thus far, only 3 Ncps Ncp-A1, A2 and M1 were isolated as pure natural products from cyanobacteria strains. Again the word ‘identified’ is not appropriate here.
  • The terminology ‘variant’ is not really best. I suggest using ‘analogues’ ,’structures’, ‘natural products’ or ‘Ncp’. Please modify everywhere.
  • I suggest clarifying this sentence. In our work, ten different Ncps were detected by LCMS in the CCNP1411 extract. These include five cyclic Ncps, four linear Ncps aldehyde and one linear hexapeptide Ncp. You might also add here the name/code of the compounds. And clarify which one are new.
  • This sentence is not clear. Something is missing here. I understand only now….after reading many times that the Ncp-E1,-E1L, -E2 and -E2L are part of the ten Ncps you are mentioning before. Is this correct? And what I also finally understand is that these 4 metabolites and new. Is this correct? If that’s the case, please clarify the sentence. And what about Ncp-E3 and -E4?
  • this sentence is not clear. What do you mean with b ions. Do you mean fragments? It’s not clear. The example used is not clear.
  • Fragmentation spectrum? Do you mean MSMS spectrum of Ncp-A1? You can see fragment in a MS1 or MS2 spectra.
  • How did you do structure analysis? Did you interpret MS fragments manually? Or did you use a database with a match factor?
  • Figure 7,8 and 9. Why do you show a structure in figure 9 and not in the other figures? I would be also nice to have some annotation on these spectra. Show some relevant fragments for example that helped in structure elucidation and to support your discussion p8.
  • I suggest replacing (without methyl group) with (instead MePro)
  • ‘…two structural isomers’. Which one are you talking about?
  • ** Ncp-E3 and then in the legend, you wrote Ncp-E6?
  • Start sentence with ‘However, the two ions m/z 101 and 129, which are characteristic of Gln, suggested the presence of this amino acid in Ncp-E1 and -E2.

Material and methods.

  • You identified the presence of new Ncps in the extract. You analyzed the data with Analyst Software. How do you analyze results and what do you get? Do you search in a DB? Do you get results as match factor? Please clarify this point with 1-2 sentences. I suggest also to clarify this point in page 8.
  • These new Ncps were identified by LCM in crude extract. These are kind of preliminary information. Don’t you think that an interesting outlook would be to isolate these metabolites and investigate their activity? I would suggest to add this perspective into the conclusion. I hope you don’t do that work just to publish something but to help move science go forward.

Conclusion.

  • You might be more precise.

All the best,

Author Response

We appreciated the effort of the Reviewer that it took to prepare such a constructive and detailed review. It significantly helped to improve the quality of the submitted work.

The paper describes a cyanobacteria strain CCNP1411 producing new nostocyclopeptides. The DNA analysis of this strain has been done and Ncps metabolites were identified by LC-MS/MS in the crude extract. I cannot judge on the molecular biology part, but can give you some comment on the other part, hoping this can help improve the manuscript:

Here are my comments:

Experimental design

  • I don’t see the relevance of determining the relative amounts of Ncps in different extracts? I would rather try to increase the total yield in order to scale-up the extraction and isolation of these metabolites?!? Detecting new compounds are ok, but these are only preliminary results. The most important is still to isolate compounds and test. I guess as I read your last paragraph L350, you would agree, isn’t it? Your experimental design (experiment L301) doesn’t match your overall goal. If this is not relevant, I suggest removing that part.

We totally agree with the Reviewer. The interesting thing and the goal of our further studies is to isolate the Ncp analogues (especially the new ones) from the strain CCNP1411 and test their activity (and structure-activity relationship) against different targets. However, for this purpose, we need more material to be collected. The high-volume cultures are currently grown. The goal of the current work was to characterize the Nostoc edaphicum CCNP1411 as a new producer of these bioactive (as documented in other studies) nonribosomal peptides by (1) sequencing Nostoc genome, (2) characterization of the ncp gene cluster and (3) gaining knowledge about the products of the gene cluster (including determination of the relative amounts of the compounds). So, we think that the experiments performed and described in section 2.4 are strictly related to the goal of the work. As it was explained in the text, we extracted the peptides with different solvents and under different conditions just to be sure that the high amount of the linear Ncps detected in Nostoc CCNP1411 is not an artefact. This was necessary for the reliability of the presented results, as in previous studies on Ncp in Nostoc, almost exclusively, the cyclic forms were described. As it was presented in the Discussion, the relative quantities of the individual peptide analogues, to some extent, corresponded to the results of genetic analysis.

General draft.

  • Please be rigorous with terminology used. The terminology ‘isolated’, ‘detected’ or ‘discovered’ should be used in a clear way. When you mentioning a compound, it is not always clear if you are talking about a compound that was detected (compound detected by LC-MS in a crude extract) or isolated (you have a pure compound in a vial and you can acquire NMR and test for biological activity). This should be clarified everywhere in the manuscript (from abstract, in figures and unto the conclusion).

I give you one example: in figure 1 where you show Ncp-E1 and Ncp-E1-L. You mention that these were isolated from CCNP 1411. To my understanding, these were only detected by LC-MS in a crude extract.

We totally agree with the Reviewer. Since crude extracts were analyzed by LC-MS/MS, the words “isolated” and “identified” should not be used in reference to our results. The respective corrections have been introduced in the manuscript.

Abstract.

  • Detected as naturally produced cyanometabolites. What do you mean? Do you mean that only three Ncps (Ncp-A1, Ncp-A2 and Ncp-M) were isolated as pure natural products from cyanobacteria? It would be nice to have this clearly written somewhere.

As it was mentioned in Abstract, Introduction and Results and Discussion, so far, only three structural analogues, namely Ncp-A1, Ncp-A2 and Ncp-M1, and their linear forms were identified in Nostoc strain (the only producer of the peptides). To make it clear: (1) in the abstract, the names of the three analogues were added, (2) in Introduction, the following part was rephrased:

“Thus far, only three analogues of the compounds and their linear forms have been discovered. This includes Ncp-A1 and Ncp-A2 detected in Nostoc sp. ATCC53789….”.

In abstract and section 2.3 we used the phrase “naturally produced cyanometabolites”, because some Ncps analogues (detected in our work as naturally produced peptides) were previously chemically synthesized, but not found in the cyanobacterium. In case of Ncp-A3, the MePhe-containing analogues was obtained in culture supplemented with MePhe, so Ncp-A3 also cannot be classified to naturally produced nostocyclopeptides.

  • Please replace the ‘four new variants’ with ‘four new analogues’ or ‘four new Ncps’. And it would be nice to add their code? Which are the new Ncps you have identified in that strain. Please mention as well if they were detected as trace-amount. Information should be clearly presented to the reader.

The sentence was rephrased as follows: “Besides the known peptides, Ncp-A1 and Ncp-A2, six new analogues were detected. This includes Ncp-E1 and E2 and their linear forms (Ncp-E1-L and E2-L), a cyclic Ncp-E3 and a linear Ncp-E4-L.”

In Table 3, the new analogues are marked in bold and the peptides detected in trace amounts are marked with [T].

Introduction.

  • This sentence is not clear to me. You say that nostocyclopeptides were identified only in Nostoc sp. then you mention only Ncp_A1, Ncp_A2, and Ncp_M1. What about Ncp_B1-B4 for example? Were these compounds never isolated and only identified by LCMS and/or through the gene clusters analysis? The term ‘discovered’ is also not clear here. Maybe you wanted to say that only three Ncp namely Ncp-A1, -A2 and -M were isolated from Nostoc All other analogues were identified by LCMS…etc. Information should be clear!!

It is mentioned in the Introduction that Ncp are produced exclusively by the genus Nostoc. Then, the few examples of Ncps and Ncps-producing Nostoc strains have been described in Introduction and in section 2.3. Also later in the text, the previously identified analogues were mentioned. To the best of our knowledge, no other Ncps were detected. To make the text clear, one sentence was removed from introduction (“Ncps constitute a small class of nonribosomal peptides”). It could distract attention of the reviewer from the thought we wanted to express. The information on nonribosomal biosynthesis of Ncps is given further in the text.

NcpB1-NcpB4 (without dash) are modules in the ncp gene cluster, not nostocyclopeptide analogues.

  • You review the activity of nostocyclopeptide. Is this relevant in this paper? This is not a review, but an introduction of your work where you are even not testing any compound. This paragraph could be shortened. I would suggest to keep on the red line.

As suggested, the text on the bioactivity of Ncps was shortened to the minimum (see below) required to justify the interest in this class of peptides as potential drugs.

“The activity of Ncps has been explored [13] and their potential as antitoxins, inhibiting the transport of hepatotoxic microcystin-LR and nodularin into the rat hepatocytes through the organic anion transporter polypeptides OATP1B1/1B3 was revealed [21]. As OATP1B3 is over expressed in some malignant tumors (e.g. colon carcinomas) [22], Ncps, as inhibitors of this transporter protein, are suggested to be promising lead compounds for new drug development”.

  • You mention that you report in this paper the production of nostocyclopeptides by Nostoc edaphicum. We still don’t understand why you are doing this work? what is your strategy behind? Are you studying these microbes? Are you looking for new cynobacteria? Are you sequencing all strains you isolate? Your strategy is not clear to me. I would appreciate 1-2 sentences about your strategy so the reader can understand why you are now reporting this strain. Maybe you could sell it in the previous paragraph (this is only a suggestion based on my understanding): nostocyclopeptides are displaying interesting activities such as xyz to mention just a few -> our goal was to isolated cyanobacteria from the Baltic sea -> Nostoc edaphicum was identified….

The last part of Introduction, presenting the aim of the study was modified, according to suggestions of Reviewer 3.

“In our previous studies, Nostoc edaphicum CCNP1411 from the Baltic Sea was found to be a rich source of cyanopeptolins, the nonribosomal peptides with potent inhibitory activity against serine proteases [6]. In the current work, the potential of the strain to produce other bioactive metabolites was explored. The whole genome sequence of N. edaphicum CCNP1411 has been determined, and the nostocyclopeptide biosynthetic gene cluster has been identified in the strain and characterized in silico for the first time. Furthermore, the new products of the Ncp gene cluster have been detected and their structures have been characterized by LC-MS/MS”.

  • Is Nostoc edaphicum a new strain? or is that strain already known and the Ncps gene cluster has never been investigated?

A sentence referring to the previous studies on Nostoc edaphicum CCNP1411 was added (see above). N. edaphicum species was previously described by Kondrateva [1962] and Komarek [2013]. These references were added to the text, together with the brief morphological description of the strain. The ncp gene cluster in N. edaphicum – the new producer of the nostocyclopeptides - was analyzed here for the first time.

  • These last sentences should be part of the conclusion. This has nothing to do in the introduction. In the introduction, you should only give information to help the reader understand why you did this work, why is this important.

That’s right. The two sentences were removed.

Figures.

  • Figure 1. Please move into results. These structures have nothing to do in introduction. Introduction is not a resumé! Structures should go in Results.

   As in the introduction a basic information on nostocyclopeptide structure (characteristic features) is given, we though it is indispensable to present the example structure of the peptide close to the description, i.e. at the end of the Introduction. In the last version of the manuscript (Figure 1), we show only one Ncp structure and a photo of the strain, as suggested by Reviewer 1.

  • The legend of that figure mention that these two compounds were isolated from CCNP1411. However, in method, I didn’t find the description of the isolation. Please be clear.

Of course, in this case, the word “detected” (not “isolated”) should be used. The correction was made.

Results and discussion.

  • L233-L272. Could you please double check the structure elucidation part and make sure the info is clearly presented and precise. The information is often confusing.

The structure elucidation and the text in section 2.3 was double checked. The additional changes introduced include:

  • Removal of the first sentence;
  • References to Figures and the Supplementary materials were added;
  • In Figure 8, some important (b/y) fragment ions are highlighted, as suggested by the Reviewer. As the same fragments occur in some other spectra, they are highlighted only in one figure (structure).
  • You are talking about data in suppl info. So please, mention it.

The references to Supplementary materials were added. In the supplementary materials, the results of Ncps analysis in extracts of different pH was additionally presented as Figure S8 (missed in the previous version). 

  • As you are discussing some fragments, would it be possible to highlight these typical fragments in the spectra (show 1 or 2 most relevant fragments next to peak). It would help the reader follow your discussion.

As the detailed description of the ion structures is given in Figure captions, we think that there is no need to double the information in the same figure. However, to meet the demands of the Reviewer, in Figure 8, some important fragment ions are marked in the structure – as an example.

  • Could you please clarify which compounds are new? In the abstract you are talking about 4 new Ncps and in table 3, I have a list of 10 Ncps.

In Table 3, we marked in bold letters the new nostocyclopeptides detected in the study as compounds naturally produced by Nostoc CCNP1411.

  • Moreover, which Ncps were present in trace amount? This should be clear from the beginning.

In Table 3, the Ncps produced in trace amounts were marked with [T].

  • Table 3. Structures. At the beginning of paragraph, would it be possible to highlight the difference between structures, so the reader can quickly understand it. I had to look carefully at the table to see that there is only a variation of 2 aa between the structures.

In section 2.3, a sentence was modified as follows:

“In our work, ten Ncps, differing mainly in positions 4 and 7, were detected by LC-MS/MS in the N. edaphicum CCNP1411 cell extract (Table 3, Figure 1, 8-9, Figure S1-S7)”.

In addition, in Table 3, the two most variable residues are marked in blue.

  • L219-220. Thus far, only three naturally produced Ncp variants …. This sentence is not clear. I would suggest to modify as follow: Thus far, only 3 Ncps Ncp-A1, A2 and M1 were isolated as pure natural products from cyanobacteria strains. Again the word ‘identified’ is not appropriate here.

The text was changed as suggested: “Thus far, only three Ncps, Ncp-A1, A2 and M1, and their linear aldehydes were isolated as pure natural products of Nostoc strains”

  • The terminology ‘variant’ is not really best. I suggest using ‘analogues’ ,’structures’, ‘natural products’ or ‘Ncp’. Please modify everywhere.

In the whole text, as suggested, „variants” was changed into „analogues”.

  • I suggest clarifying this sentence. In our work, ten different Ncps were detected by LCMS in the CCNP1411 extract. These include five cyclic Ncps, four linear Ncps aldehyde and one linear hexapeptide Ncp. You might also add here the name/code of the compounds. And clarify which one are new.

The text was changed as follows:

‘In our work, ten Ncps, differing mainly in positions 4 and 7, were detected by LC-MS/MS in the N. edaphicum CCNP1411 cell extract (Table 3, Figure 1, 8-9, Figure S1-S7). These include five cyclic structures, four linear Ncp aldehydes, and one linear hexapeptide Ncp. The six nostocyclopeptides marked in Table 3 in bold (Ncp-E1, Ncp-E1-L, Ncp-E2, Ncp-E2-L, Ncp-E3 and Ncp-E4-L) were found to be naturally produced by Nostoc for the first time”.

  • This sentence is not clear. Something is missing here. I understand only now….after reading many times that the Ncp-E1,-E1L, -E2 and -E2L are part of the ten Ncps you are mentioning before. Is this correct? And what I also finally understand is that these 4 metabolites are new. Is this correct? If that’s the case, please clarify the sentence. And what about Ncp-E3 and -E4?

See the modified text above.

  • this sentence is not clear. What do you mean with b ions. Do you mean fragments? It’s not clear. The example used is not clear.

The sentence was rewarded as follows:

“The process of de novo structure elucidation was performed manually, based mainly on a series of b and y fragment ions produced by cleavage of the peptide bonds (Figure 8), and on the presence of immonium ions (e.g. m/z 70 for Pro, 84 for MePro, 136 for Tyr) in the product ion mass spectra of the peptides.”

  • Fragmentation spectrum? Do you mean MSMS spectrum of Ncp-A1? You can see fragment in a MS1 or MS2 spectra.

Corrected. Fragmentation spectrum changed into ”product ion mass spectrum”

  • How did you do structure analysis? Did you interpret MS fragments manually? Or did you use a database with a match factor?

The structure elucidation was performed manually, based on product ion mass spectra with fragment ions, as described in the work (section 2.3).

  • Figure 7,8 and 9. Why do you show a structure in figure 9 and not in the other figures? I would be also nice to have some annotation on these spectra. Show some relevant fragments for example that helped in structure elucidation and to support your discussion p8.

Structure of Ncp-E1-L was transferred from Figure 1 to Figure 8, which presents spectrum of Ncp-E1-L. Because of the reason explained above (i.e. the text in Introduction), the structure of Ncp-E1, which corresponds to the spectrum presented in Figure 7, was left as Figure 1.

As we explained above, if we add the annotation on the spectra, the same information would be double in the figures. The detailed description of the fragment ions is given in each figure caption.

For higher clarity of the text and presented results, some important ions (b/y) are marked in Figure 8.

  • I suggest replacing (without methyl group) with (instead MePro)

Changed as suggested.

  • ‘…two structural isomers’. Which one are you talking about?
